# Detection of SARS-CoV-2–Specific Antibodies in Human Breast Milk and Their Neutralizing Capacity after COVID-19 Vaccination: A Systematic Review

**DOI:** 10.3390/ijms24032957

**Published:** 2023-02-03

**Authors:** Vicky Nicolaidou, Rafaela Georgiou, Maria Christofidou, Kyriacos Felekkis, Myrtani Pieri, Christos Papaneophytou

**Affiliations:** Department of Life Sciences, School of Life and Health Sciences, University of Nicosia, Nicosia 2417, Cyprus

**Keywords:** SARS-CoV-2, COVID-19, breast milk, antibodies, vaccines, lactating

## Abstract

SARS-CoV-2 is the virus that causes the infectious disease known as Corona Virus Disease 2019 (COVID-19). The severe impact of the virus on humans is undeniable, which is why effective vaccines were highly anticipated. As of 12 January 2022, nine vaccines have obtained Emergency Use Listing by the World Health Organization (WHO), and four of these are approved or authorized by the Centers for Disease Control and Prevention (CDC) in the United States. The initial clinical trials studying COVID-19 vaccine efficacy excluded pregnant and lactating individuals, meaning that data on the effects of the vaccine on breast milk were lacking. Until today, none of the authorized vaccines have been approved for use in individuals under six months. During the first months of life, babies do not produce their own antibodies; therefore, antibodies contained in their mothers’ breastmilk are a critical protective mechanism. Several studies have shown the presence of SARS-CoV-2 antibodies in the breast milk of women who have been vaccinated or had been naturally infected. However, whether these are protective is still unclear. Additionally, research on the BNT162b2 mRNA vaccine developed by Pfizer-BioNTech and the mRNA-1273 vaccine developed by Moderna suggests that these vaccines do not release significant amounts, if any, of mRNA into breast milk. Hence, there is no evidence that vaccination of the mother poses any risk to the breastfed infant, while the antibodies present in breast milk may offer protection against the virus. The primary objective of this systematic review is to summarize the current understanding of the presence of immunoglobulins in human milk that are elicited by SARS-CoV-2 vaccines and to evaluate their ability to neutralize the virus. Additionally, we aim to quantify the side effects experienced by lactating mothers who have been vaccinated, as well as the potential for adverse effects in their infants. This study is critical because it can help inform decision-making by examining the current understanding of antibody secretion in breastmilk. This is particularly important because, although the virus tends to be less severe in younger individuals, infants who contract the disease are at a higher risk of requiring hospitalization compared to older children.

## 1. Introduction

The severe acute respiratory syndrome coronavirus 2 (SARS-CoV-2), which causes COVID-19, has profoundly impacted the world, leading the scientific community to urgently seek answers and develop treatment and protection strategies [1]. As a result of the rapid spread and burden of COVID-19, the regulatory authorities authorized vaccines against SARS-CoV-2 for emergency use [2]. The range of platform technologies used includes traditional methods such as whole live-attenuated and inactivated virus vaccines, protein subunit vaccines, and virus-like particles (VLPs), as well as novel technologies that have never been used for licensed human vaccines, including nucleic acid vaccines (mRNA and DNA), replicating and nonreplicating viral and bacterial vectors, and modified antigen-presenting cells and T cells [3]. Six vaccines that target COVID-19 have been approved for use in Europe by the European Medicines Agency (EMA), including two that use mRNA technology (BNT162b2 mRNA by Pfizer-BioNTech and mRNA-1273 by Moderna) and two that use recombinant adenovirus technology (ChAdOx1 nCoV-19 by Oxford/AstraZeneca and Ad26.COV2.S by Johnson & Johnson). However, the initial clinical trials for these vaccines did not include pregnant and lactating women [4,5]. This initial exclusion of this population from early vaccination studies resulted in a lack of data for supporting evidence-based recommendations, leading to the exclusion of pregnant women from vaccination rollout plans [6]. This makes the generation of further evidence very challenging, a situation that was recently termed a “perpetuated cycle of exclusion” [7]. However, acknowledging the pandemic emergency and the imperative to ensure the access of pregnant or lactating women to vaccine-mediated protection against COVID-19, various public health organizations recommend the vaccination of these subpopulations (see [8]; there are references cited therein).

Since the rollout of the vaccines, several studies led by us [9] and others (reviewed by [10]) have demonstrated the presence of antibodies in the breast milk of vaccinated lactating mothers. Some studies have addressed the resilience of these antibodies throughout simulated digestion systems [9], the neutralization capacity of antibodies in breastmilk [11], as well as the overall safety of the vaccine in lactating mothers and infants [12].

Breastmilk offers some immunological protection to infants against infectious diseases, including COVID-19, due to the presence of immune-stimulating factors such as antibodies, cytokines, oligosaccharides, and nucleic acids (reviewed in [13]). The World Health Organization (WHO) and various scientific societies around the world recommend that new mothers exclusively breastfeed their infants for the first six months and then continue to breastfeed while also introducing solid foods for at least two years [14,15]. IgA is the most abundant immunoglobulin (~90%) in human milk, and it is essential in conferring mucosal immunity, while IgM and IgG are also present at lower concentrations (~8% and ~2%, respectively) [16]. IgG is found at the lowest concentration because it usually originates from the blood [16]. Both IgA and IgM arise from the mammary gland—specifically, mucosa-associated lymphatic tissue (MALT)—and they are secreted in a polymeric form attached to a j-chain and secretory component proteins [17]. IgA provides immunity by inhibiting the binding of pathogens to the host cells in the mucous membranes of the respiratory and gastrointestinal tract [18]. A recent study by Sterlin et al. [19] demonstrated that IgA levels specific to the receptor-binding domain of the virus spike protein are proportional to their neutralizing ability. Thus, the induced levels of IgA titers through vaccination against SARS-CoV-2 are crucial in understanding the passive transmission and protection of the infant.

The specific vaccination route is essential for immunity generation. mRNA vaccines are administered through the intramuscular route, which elicits a systematic immune response. While mucosal immunity is typically achieved through the intranasal or oral vaccination route, intramuscular vaccination can also result in the transfer of IgA to breast milk, as seen with the influenza vaccine. It is yet to be determined whether the SARS-CoV-2 vaccines produce IgA in the form of MALT-derived secretory IgA or its monomeric form from the blood [20]. IgM is normally found in its pentameric form as sIgM, and it is involved in the activation of the complement system and generates agglutination in regard to pathogens [11].

Despite studies showing that the benefits of vaccination outweigh the risks for both humans and animals, many pregnant or lactating women still reject the COVID-19 vaccine out of fear of harming themselves or their children [21]. Therefore, accurate information about this population is critical in promoting vaccination and preventing women from prematurely ending their breastfeeding journey [22].

For many respiratory infectious diseases, pregnant women and neonates are two high-risk populations that suffer disproportionate rates of morbidity and mortality [23]. Importantly, infants have an immature immune system and rely on the transfer of antibodies and maternal immune cells via breastfeeding to provide them with immunity [24]. While it is well known that infants receive passive immunity from milk antibodies after vaccination, there is still hesitancy among lactating mothers [25]. This is mainly due to a lack of knowledge about the effects of mRNA-based vaccines on nursing infants, as lactating mothers were excluded from the initial clinical trials of mRNA vaccination [25].

As of the writing of this manuscript, vaccines for SARS-CoV-2 have not yet been approved for use in children under six months of age, leaving this group vulnerable to potential infection. The only protection for these young infants is through the passive immunity provided by the breast milk of lactating mothers, or in the case of pregnant women, through the placenta to the unborn fetus [26]. Fortunately, children infected with SARS-CoV-2 have generally had a positive clinical presentation. However, there have been cases of infected infants where complications such as mechanical ventilation, cardiac and liver dysfunction, and even death have been reported. The immune system of infants is still developing during the first six months, leaving them more vulnerable to infections compared to older children [27].

The purpose of this systematic review is to identify and summarize the current understanding of the presence of immunoglobulins in human milk that have been elicited by vaccines against SARS-CoV-2 and to evaluate their ability to neutralize the virus. We also aim to quantify the side effects experienced by individuals who have been vaccinated and the potential for adverse effects in their infants.

## 2. Methods

### 2.1. Study Search

A systematic search was conducted in the PubMed and Scopus databases to identify all relevant scientific literature on the topic. The search covered all fields in the databases, as this was the first systematic effort to gather such information. Data collection was carried out from December 2020 to May 2022 using the following search string: (((ALL = (COVID 19)) AND ALL = (vaccine)) AND ALL = (breast milk)) AND ALL = (lactating). The newly established LitCOVID database (https://doi.org/10.1093/nar/gkaa952 (accessed on 31 May 2022) was also checked to identify any potential additional literature.

### 2.2. Inclusion and Exclusion Criteria

The selection process for this review included two criteria for inclusion: (1) women vaccinated against SARS-CoV-2 and (2) lactating individuals. Two exclusion criteria were also set: (1) women infected with SARS-CoV-2 and (2) reports not written in English. Additionally, the research was limited to publications that were either freely available or accessible through institutional subscriptions. A timeframe of December 2020 to May 2022 was also established, and pre-prints were not included in the study. All studies included women aged 18 years or older, even though no age limit was set during the search.

### 2.3. Study Selection

The selection of articles was carried out using the general principles of the PRISMA 2020 statement for the identification of studies via databases and registries [28]. The specifics of the final article selection are shown in the PRISMA 2020 flow diagram in Figure 1. The references were then all transferred to the EndNote (version X8) platform (Clarivate Analytics, Philadelphia, PA, USA). Duplicates present between the different databases were removed during the selection process of articles (Figure 1). Three authors (V.N., R.G. and M.P.) independently screened the titles, abstracts, and full texts of articles for eligibility, assessed their generalizability, and collected data. If necessary, discrepancies were resolved through discussion with a fourth author (C.P.).

## 3. Results and Discussion

### 3.1. Antibody Levels against SARS-CoV-2 in Human Milk Following Vaccination

Herein, we summarize the current knowledge regarding the presence and types of SARS-CoV-2-specific immunoglobins in human milk after vaccination, their ability to neutralize the virus, as well as putative side-effects experienced by the lactating mother and infant/baby. Eighty-five publications were identified from the database search (PubMed and Scopus), of which thirteen were removed for being duplicate records. Of the 72 remaining records, 42 were excluded based on the screening of their titles. From the 30 records that remained for abstract screening, 2 were excluded for a lack of original research and 5 were excluded for not including relevant information. Eventually, 23 publications were selected for full-text reviewing, and 1 study was excluded for not having relevant research. In the end, 22 papers were identified and chosen for the analysis (Figure 1).

A total of 825 participants took part in the 22 studies. Table 1 contains brief summaries of the characteristics, methodologies, and results of the included studies. The Enzyme-linked immunosorbent assay (ELISA) was employed in the majority of these studies (*n* = 20; the method used is not mentioned in one study) for the identification of anti-SARS-CoV-2 antibodies in serum and human milk samples. Furthermore, in six of the selected studies [29,30,31,32,33,34], a pseudovirus neutralization (functional) assay was carried out to evaluate the SARS-CoV-2 neutralization capacity of breast milk (Table 1). In one study [26], the neutralization capacity of antibodies was determined using a V-PLEX SARS-CoV-2 Panel 6 multiplex assay (Mecosle Discovery). In seven other studies [12,20,26,35,36,37,38,39], breastmilk samples were tested by ELISA for receptor-binding domain (RBD)-specific IgG and IgA that could inhibit the binding of the spike protein with the ACE-2 receptor (Table 1).

Several limitations were identified in the included studies, the most significant of which was the small sample size (ranging from 7 to 110 individuals). Furthermore, most studies (15 out of 22) included only healthcare professionals, which did not provide a diverse range of subjects [39,40]. Additionally, participants were required to collect and store their own breast milk samples, which could have affected the quality and preservation of the samples. Furthermore, the different time points at which milk and serum were collected in the studies led to inconsistent results. Finally, not all studies included functional assays for assessing the effectiveness of SARS-CoV-2 neutralization in human milk after vaccination.

**Table 1 ijms-24-02957-t001:** Study characteristics and main findings.

n	Vaccine Type	Sampling Time	Detection Method	Main Findings	Ref.
23 (HCP ^a^)	mRNA-1273 (*n* = 2) and BNT162b2 (*n* = 21)	Milk & blood samples: Prior to vaccination 10 days after 1st & 2nd dose and 4–10 weeks after second dose	ELISA & Neutralizing antibodies (anti-RBD ^b^)	Anti-spike Abs in circulation: 73% IgA+ IgG+ IgM+, 8.69% IgG+ IgM+ & 4.34% IgA+ IgG+ 95.65% had anti-spike Abs in breastmilk, with 4.34% IgA+ IgG+ IgM+, 13% IgA+, & 73% IgA+ /IgG+ Anti-spike SIgA ^c^ in 70% of milk samples	[20]
84 (HCP)	BNT162b2	Milk samples: Prior to vaccination Weekly for six weeks after the first dose	ELISA	IgA increased two weeks after the first dose, with 61.8% being positive; at week four, this increased to 86.1%, and at week six, it decreased to 65.7% IgG increased four weeks post-vaccination to 91.7% and peaked at 97% at weeks 5–6	[11]
35 (HCP)	BNT162b2	Milk & blood samples: Prior to vaccination 1, 3, 7, 14, & 21 days post-vaccination with each dose	ELISA & Neutralizing Antibodies	21 days after: First dose: 74.2% IgG, 100% IgA, & 83.9% IgM, SARS-CoV-2 spike RBD Second dose: 100% IgG1 & IgA, 88.6% IgM Moderate correlation of serum and milk neutralizing Abs 7 days after second dose	[38]
29	BNT162b2 (*n* = 26) mRNA-1273 (*n* = 3)	Milk & blood samples: Prior to vaccination 1, 3, & 6 months after vaccination	ELISA & Neutralizing Antibodies	IgG Abs peaked at 1 month post-vaccination and were detected 6 months post-vaccination IgMs were undetectable at 6 months post-vaccination. IgA Abs were detected in 50% & 25.91% of milk samples 1 and 3 months post-vaccination, respectively; at 6 months post-vaccination, they reached baseline Strong correlation between milk & serum IgGs Neutralization Abs were detected in 83.3%, 70.4%, and 25.0% of milk samples at 1, 3, and 6 months post-vaccination, respectively	[33]
28 +18 with prior infection	BNT162b2	Milk samples: Prior to vaccination 3, 5, 7, 9, 11, and 15–17 days after first & second dose	ELISA	IgAs: high levels 2 weeks after first dose. Five days after second dose, this began to increase and peaked at 1 week. First dose: 15 days post-vaccination, 73% of participants had milk conversion. Second dose: 96% of participants reached milk conversion IgAs in milk were determined 70 days after vaccination and infection	[41]
30	BNT162b2 (*n* = 20) mRNA-1273 (*n* = 10)	Milk samples: Prior to vaccination 0–53 days after administration of the first dose	ELISA & Neutralizing assay	Vaccination induced the production of anti-RBD IgG Abs in breastmilk Anti-RBD IgG and anti-RBD IgA may be transferred from mother to breastfed infant	[26]
26	BNT162b2	Milk samples: Prior to vaccination 3, 5, 7, 9, 11, 13, & 15 days after each dose	ELISA	IgA increased 5–7 days after first dose; 12% increase per day, reaching a threefold increase compared to pre-vaccination levels 15 days after first dose, IgA levels decreased by 43% and stabilized at 50% of the peak level 2.3-fold increase in IgA after second dose IgA levels declined by 33% after a week of full vaccination	[27]
31	mRNA-1273 (*n* = 15) BNT162b2 (*n* = 16)	Milk samples: V_0_ =Prior to vaccination V_1_ = Second dose time V_2_ = 2–6 weeks after the second dose	ELISA & Pseudo virus neutralizing assay	Serum samples: an increase in all isotypes was observed from V_0_–V_1_. IgG levels increased from the V_1_–V_2_ timepoint; IgG was the dominant serum Ab. IgM & IgA increased after the first dose, but after the second dose, no significant Ab boosting was observed Milk samples: a strong induction of IgG, IgA, and IgM was observed after both doses. IgA and IgM did not increase after the second dose. IgG had a boost in milk samples; IgG1 RBD increased significantly from V_0_–V_2_. Anti-RBD IgA & IgM levels did not increase after either dose	[29]
16 + 6 with prior infection	mRNA-1273 (*n* = 5) BNT162b2 (*n* = 11)	Milk & serum samples: Before each dose 2–8 weeks after second dose Infant cord blood at delivery	RBD-ELISA & Pseudo virus neutralizing assay	Serum RBD-IgG and -IgA titers were ~16-fold and 5.4-fold higher, respectively, after vaccination compared to natural infection Milk RBD IgG and IgA titers were 2-fold and 77-fold lower, respectively, after natural infection compared to vaccination. Binding and neutralizing antibodies were detected in infant cord blood	[30]
61	BNT162b2	Blood and breastmilk samples were collected after delivery Blood spots & saliva samples from infants at 30, 90, and 150 min after breastfeeding	RBD ELISA & Pseudo virus neutralizing assay	IgG antibodies were detected in all maternal samples Significantly positive correlation between serum & breastmilk SARS-CoV-2 IgG levels 38.3% of milk samples exhibited neutralizing activity SIgAs^c^ were detected in 15% of breastmilk samples	[31]
30 (HCP)	mRNA-1273 (*n* = 12) BNT162b2 (*n* = 18)	Milk samples: Prior to vaccination 18 days after first dose 18 & 90 days after second dose	ELISA & Neutralization assay	IgG increased 18 days after the first dose and further after the second dose. IgG levels peaked at 90 days post-vaccination IgA levels increased 18 days after the first dose, with no increase noted after the second dose Neutralizing activity of 20 milk samples: 60% increased 18 days after the first dose & 85% remained elevated 18 days after the second dose No significant differences in Abs production between the two vaccines	[32]
10 (HCP)	BNT162b2	Milk & serum samples: Pre-pandemic controls 7 and 14 days after first & second doses	ELISA & Spike-bearing pseudovirus neutralization assay	IgG and IgA titers in serum and milk increased significantly 14 days after the first dose. IgA in serum & breastmilk peaked 7 days after the second dose, suggesting a delay in accumulation compared to IgG. Serum samples: Ab response was dominated by IgG The IgG: IgA ratio in milk indicated that the IgA response was dominated by IgA, but at 7 and 14 days after the second dose, IgG levels increased All milk samples were observed to have a neutralization capacity 7 days after the second dose	[34]
110 (HCP)	BNT162b2 (*n* = 70) mRNA-1273 (*n* = 20) ChAdOx1-S ^d^ (*n* = 20)	Milk & serum samples: BNT162b2 & miRNA-1273: 30 days after the second dose ChAdOx1-S: 30 days after first dose	IgG & IgA: ELISA Anti-S1 IgG: CLIA ^e^	Higher IgG & IgA levels in both serum & milk samples obtained from lactating mothers vaccinated with BNT162b2 & miRNA-1273 compared to ChAdOx1-S. mRNA vaccines: Significant correlation between serum IgG and IgA levels All vaccinated mothers had serum anti-S1 Abs ^f^ Higher levels of Abs were detected in the mRNA vaccine category	[35]
42 (HCP)	mRNA-1273 (*n* = 1) BNT162b2 (*n* = 36) AZD1222 (*n* = 4) AZD1222 & BNT162b2 (*n* = 1)	Milk & serum samples: 20–30 days after second dose 1–2 months after second dose 3–4 months after second dose	ELISA & Neutralizing Antibodies	Anti-SARS-CoV-2 IgGs were detected in all milk and serum samples Serum levels of IgG were higher than those in milk samples Anti-SARS-CoV-2 IgA Abs were not detected in milk samples	[36]
50	mRNA-1273 (*n* = 21) BNT162b2 (*n* = 27)	Milk & serum samples: Prior to vaccination Before second dose 4–10 weeks after second dose	Pylon 3D automated immunoassay and ELISA	IgG increased after the first dose and significantly increased after the second dose Anti-SARS-CoV-2 RDB- IgAs reached the highest levels after the first dose but did not significantly increase after the second dose IgM levels increased significantly after the first dose and did not change after the second dose No significant difference in Abs levels between the two types of vaccines	[12]
98 & 24 controls (HCP)	mRNA-1273 (*n* = 6) BNT162b2 (*n* = 92)	Milk & serum samples:14 days after second dose	ELISA & Neutralizing antibodies	Serum: Anti-SARS-CoV-2 S1 IgM detected in 22.5% of samples & IgG detected in all samples Control serum samples were negative for both IgG & IgM anti-SARS-CoV-2 N and SARS-CoV-2 spike RBD SARS-CoV-2 RBD-S1 IgG Abs were detected in all milk samples obtained from vaccinated mothers Anti-SARS-CoV-2 RBD-S1 IgG levels in milk samples were lower than those in serum samples but higher compared to the controls 89% of the vaccinated population had anti-SARS-CoV-2 S1 IgA Anti-SARS-CoV-2 RDB-S1 IgMs were not detected in milk samples IgMs were detected in human milk samples	[37]
7 (HCP)	mRNA-1273 (*n* = N.A ^g^) BNT162b2 (*n* = N.A)	Milk samples: (i) prior to vaccination; (ii) 1, 4, 7, 11, & 14 days after first dose; (iii) 1 day before the second dose; (iv) 1, 4, 7, 11, 7, and 14 days after second dose; (v) 80 days after first dose	ELISA & Neutralizing antibodies	Significant increase in IgG and IgA specific to the SARS-CoV-2 viral spike protein and RBD in human milk 7 & 14 days post-vaccination At peak levels, the concentration of IgG was double that of IgA for both anti-spike & anti-RBD Igs Vaccine-induced-Ab response: IgG-dominant There was no significant difference between the two mRNA vaccines in Abs responses	[39]
14 (+10 non-lactating) (HCP)	BNT162b2	Milk and blood samples:1–3 weeks after the first & second dose	ELISA to SARS-CoV-2 trimeric spike protein	All serum samples were positive for IgM, IgA, & IgG; the latter was the dominant Ab response In breastmilk samples: IgG at 7.1% after the first dose and at 42.9% after the second dose IgA at 35.7% after the first dose, reduced to 21.4% after the second dose No IgM Abs were detected after both doses Moderate association between a more extended breastfeeding period and higher titers	[40]
33	BNT162b2	Milk and blood samples: 2 weeks after first dose 2 weeks after second dose 4 weeks after second dose	N/A	Anti-SARS-CoV-2 S1 IgG Abs were detected in serum and breast milk samples After the second dose, the IgG S1 Abs levels in human milk increased and were positively associated with serum levels	[42]
32 (HCP)	BNT162b2	Milk & blood samples: 8 ± 1 and 22 ± 2 days after first dose 7 ± 3 and 21 ± 4 days after second dose	ELISA & Neutralizing antibodies	Anti-SARS-CoV-2 IgM Abs were not detected in serum or milk samples. The highest IgA and IgG concentrations were recorded on day 29 (±3) and decreased on day 43 (±4) Strong neutralization capacity of antibodies after infection	[43]
14 (HCP)	BNT162b2	Milk samples: T_1_: Prior to vaccination T_2_: 1–3 days after first dose T3: 7–10 days after first dose T4: 3–7 days after second dose T5: 4–6 weeks after second dose	ELISA & Neutralizing antibodies	IgA & IgG were detected in the milk samples of 86% of participants 3–7 days after the second dose IgG responses did not decrease 4–6 weeks post-vaccination Anti-spike and anti-RBD IgAs were detected in milk samples, but their levels decreased 4–6 weeks post-vaccination Negligible levels of the mRNA vaccine were detected in some milk samples, but no adverse effects were reported in infants SARS-CoV-2 IgA & IgG levels were comparable; there was a co-dominance between them IgG had a more durable response than IgA, which saw a decrease 2–3 weeks after the second dose	[44]
21 (HCP)	mRNA-1273 (*n* = 7) BNT162b2 (*n* = 14)	Milk & blood samples: Pre-vaccination, 16–30 days after first dose 7–10 days after second vaccine dose	ELISA	IgA significantly increased in milk and serum after the first and second doses IgA anti-SARS-CoV-2 Abs were detected in 85% of participants after vaccination IgG tested in 10 participants; all developed anti-SARS-CoV-2 Abs after complete vaccination A significant increase between the first and second doses was observed IgA higher than IgG in human milk Blood IgG levels were detected after each dose and peaked after the second dose	[45]

^a^ HCP: healthcare professionals; ^b^ RBD: receptor binding domain; ^c^ SIgA: secretory IgA; ^d^ Only one dose was administrated due to thrombotic thrombocytopenia vaccine-induced episodes; ^e^ CLIA: Chemiluminescence Enzyme Immunoassays; ^f^ S1 Abs: Anti-spike S1 receptor-binding domain antibodies, ^g^ N.A.: non-available.

The function of the mammary MALT in the protection of the infant by producing secretory antibodies via the j-chain and proteins was highlighted by Goncalves et al. [20]. The study also focused on the secretory form of IgA (SIgA) specific to the spike protein that was detected in 87% of milk samples. This reinforced the thought that mRNA vaccines can generate immune responses via the oral and mammary mucosa. The neutralizing ability of the human milk antibodies against the virus was assessed; even though only one of the samples had neutralizing capabilities, it was subjected to purification and concentrating assays that increased its ability. This further supports the idea that breast milk IgA can provide the infant with antibodies with neutralizing abilities. The vaccine produced neutralizing abilities that seem milder than natural infection following spike SIgA, which does not elicit a response after the booster vaccine. Even though their neutralizing capabilities were the same as those in the controls, high anti-RBD IgA levels were reported in lactating individuals. A unique finding of this analysis is the high number of circulating memory B cells specific to RBD, which correlates with the IgG anti-spike levels. High anti-RBD lymphocytes are linked to prolactin, a milk-inducing hormone. This analysis summarizes that mRNA vaccine breast milk immune components result from anti-spike SIgA and T cells.

In the prospective cohort study by Golan et al. [22], lactating individuals who received mRNA vaccines were included, and their serum and breast milk were analyzed at specific times before and after each vaccine dosage. This study utilized an enzyme-linked immunoassay for milk samples and a Pylon 3D automated immunoassay for plasma samples and Polyethylene Glycol (PEGylated) proteins in human milk by ELISA. When comparing the two mRNA-based vaccines (i.e., mRNA-1273 & BNY162b2), the data showed that lactating women could experience more vaccine-induced adverse effects after the mRNA-1237 vaccine compared to the BNT-162b2. Notably, no adverse effects were reported in the infants. In the human milk analysis for proteins that were PEGylated, there were no high concentrations of PEGylated proteins noted in either dosage sample, except in one instance which could not be confirmed, but after the second dose, there was no increase. The study also highlights that vaccination with either of the two mRNA vaccines does not lead to an immune response generated by the infant. The study also exhibited a rise in anti-SARS-CoV-2 IgM and IgG isotypes in the maternal serum. IgM levels were induced at the same level after receiving both vaccine doses, but a sixfold increase in IgG after receiving both doses was reported. After the second booster shot, this rise in IgG levels demonstrated its importance in producing immunity. Anti-SARS-CoV-2 anti-RBD IgG exhibited a positive correlation between milk and serum samples; this furthers the evidence that IgG travels from the blood to breast milk. The milk anti-RBD IgA measured after the second dose administration, as with IgM, did not increase compared with the first dose. The production of antibodies is different between individuals, as 25% of the participants did not elicit any anti-RBD IgA.

Humoral and cellular immunogenic responses to mRNA vaccines were analyzed by Collier et al. [30] in various participants, including lactating women. Antibodies found in the breast milk of vaccinated individuals exhibited binding and neutralizing capabilities; IgA antibodies were found at minimal levels, with the only women with noticeable IgA being the ones who received the vaccine when pregnant. The analysis reported neutralizing antibodies in both human milk and cord blood, which further supports the hypothesis of passive maternal immunity to the infant.

Baird et al. [39] found increased IgA and IgG anti-SARS-CoV-2 for both the spike and RBD 7- and 14-days post-vaccination. The specificity of the antibodies against RBD suggests that they can inhibit the virus from entering the cells. The results point towards an IgG-dominated response generated from the vaccine, as seen in most studies. In a study conducted by Charepe et al. [40], it was found that the vaccine elicited a response that resulted in the production of both anti-spike IgA and IgG. The kinetics of these responses were similar, with the IgA response decreasing after the second dose and the IgG response increasing. Infected women were characterized by higher levels of IgA, while vaccinated women had higher levels of IgG. Additionally, a positive correlation was observed between the titers of antibodies in serum and human milk, but the levels of these antibodies were lower in milk.

In their study, Estevez-Palau et al. [42] confirmed that there are anti-SARS-CoV-2 IgG S1 antibodies in vaccinated lactating individuals’ breast milk, and following the second vaccination, the IgG levels increased and were positively correlated with the IgG serum levels. Gray et al. [29] reported a lack of IgM boosting compared to IgG titers after vaccination. Furthermore, a lack of IgA titers was observed in all women participants who received the booster shot. Even though the IgA levels did not increase after vaccination, IgG1 anti-RBD transfer rose substantially after the second dose, and high levels were transferred to the neonate through breast milk.

Jakuszko and co-workers [43] reported elevated IgG and IgA levels in serum and human milk samples, with anti-SARS-CoV-2 IgG levels increasing following the booster shot; both IgA and IgG levels were positively correlated between serum and human milk. Participants who had a history of infection with COVID-19 after vaccination had high levels of antibodies after the first dose in both serum and milk samples; due to a limitation in infected and vaccinated individuals, no clear conclusion can come from this. It is noteworthy that no adverse effects were reported in the breastfed infants from the vaccinated mothers. In agreement with the aforementioned results, Juncker et al. [27] compared the specific antibody levels and titers between lactating individuals who received the mRNA vaccine and with infected subtypes over 70 days. The research findings align with previous studies that concluded that anti-SARS-CoV-2 IgA antibodies were similar between the two groups. They state that specific antibody titers should not be compared at different time points. In previously infected individuals who received the vaccine, a higher level of IgA levels was generated due to amplification. The majority of the vaccinated participants exhibited milk conversion with less variability regarding their antibody response, as expected, since vaccines are produced to elicit a specific immune response. Moreover, the study conducted by Lechosa-Muñiz et al. [35] aimed toward the safety of the anti-SARS-CoV-2 vaccines in both infants and mothers. No adverse effects were reported in either the mother or infant. As for immunity, serum and breast milk seem to generate similar levels of IgG and IgA for the anti-spike of SARS-CoV-2. They highlighted that lactating mothers could offer their breastfed infants IgA and IgG anti-spike antibodies via breast milk.

In addition to the results of the aforementioned studies, the study by Valcarce and co-workers [45] revealed the induction of anti-SARS-CoV-2 IgA and IgG antibodies via mRNA vaccination in human breast milk, especially after the administration of the second dose. A dominance of IgA titers is reported in milk samples, even though this is not in line with other analyses where IgG dominance is reported after complete vaccination for SARS-CoV-2. The data of this study revealed that lactating mothers vaccinated with Pfizer had a higher anti-SARS-CoV-2 IgG response. Additionally, the participant in this study with the highest levels of IgA in her milk was the only individual breastfeeding two children at the time. The lactation period shows a correlation with IgA levels.

Juncker et al. [27] reported an increase in anti-SARS-CoV-2 antibodies after both COVID-19 vaccine doses. This was attributed to the different sampling timing of the study. After the second dose, a peak was reached for IgA in the milk samples, and then the levels started declining. The timing of the analysis follows the levels of antibodies in milk for up to fifteen days post-second vaccination, which afterward remain unclear if the levels continue to decline in serum samples. Anti-SARS-CoV-2 IgG increased over time, and the findings were consistent with the clinical trials of the Pfizer vaccine, with increasable IgG 21 days after the first and second vaccine. The study suggests that a more extended period of sampling human milk post-vaccination should be implemented for better results and further answers.

Specific antibodies against SARS-CoV-2 from the BNT162b2 vaccine were detected in milk samples by Low et al. [44]; three to seven days after the second dose, 86% of participants had IgA, and 100% had IgG in their milk. An analysis of the 4–6 weeks after the final vaccination revealed that SARS-CoV-2 IgG levels were sustained while the IgA levels declined, but anti-spike IgA remained positive for 90% of the participants. Negligible amounts of mRNA from the vaccine were detected in the breast milk samples. Infants were not reported to experience any adverse effects from the vaccination of their mothers. The analysis utilized a sensitive assay, phenol-chloroform extraction for RNA extraction, and double-quencher qPCR probes to increase the assay’s sensitivity. The authors reported that neglectable amounts of mRNA were found in mammary secretions; this is expected to be digested by the infant’s gut enzymes. The production of anti-RBD and anti-spike IgAs was further included after administering the second dose of the vaccine. Anti-spike and anti-RBD IgG levels were at high levels post-second dosage. When IgG and IgA were compared at a specific timepoint, specifically 3–7 days post-second vaccination, the raw data showed that SARS-CoV-2 against the spike and RBD IgG was thought to be the leading antibody compared to IgA. Despite that, when the raw values were converted to picomolar (pM), IgA and IgG were codominant. This highlighted the significance of absolute quantification when comparing IgG and IgA titers. Despite this, the IgG response was more durable after a certain period (4 to 6 weeks after the second vaccination), as IgA levels started to decline.

Narayanaswamy et al. [26] evaluated the immune responses to the mRNA vaccines against SARS-CoV-2. Along with the breast milk samples, participants provided dried blood spots. This study also included the analysis of stool samples from breastfed infants. Pre-vaccination samples were included to establish a baseline, and control samples were collected in the same way as the vaccine samples to ensure their reliability. The study utilized ELISA for the RBD of IgG and IgA and a neutralization assay to measure the presence of mRNA vaccine immune responses in human milk and the potential immunity they may provide to the breastfed infant. The results showed that RBD IgG was the dominant antibody in milk and serum samples after the second dose of the vaccine, while IgA levels were low in both. Antibodies were able to neutralize the spike protein and four other variants. The study also found high levels of interferon-gamma in milk samples, which may be linked to side effects in participants after vaccination. Infant samples were found to have anti-RBD IgG and IgA, and a small percentage of milk antibodies was found in stool samples, which may be due to gut degradation.

The prospective longitudinal study by Perez et al. [33] analyzed the presence and neutralization activity of SARS-CoV-2 antibodies in human milk and serum samples of recently vaccinated lactating individuals. The study found that vaccine-induced antibodies peaked around the one-month mark and lasted for up to six months. The dominant isotype in milk samples was IgG and remained above the baseline at the six-month mark. IgA exceeded baseline levels at three months post-vaccination but was lower than IgG. The majority of the participants exhibited neutralizing abilities at one-month post-vaccination and remained at higher baseline levels at three months. The study also analyzed the effect of the pasteurization process on the levels of IgG in human milk and its neutralizing capabilities and found no impact.

Perl et al. [11] reported that IgA anti-SARS-CoV-2 antibody levels in human milk were elevated promptly at 61.8% two weeks post-first vaccination with Pfizer and peaked at 86.1% after four weeks; at the six-week time point, a decrease was observed at 65.7%. On the other hand, anti-SARS-CoV-2 IgG antibodies increased after four weeks at 91.7%, and at weeks five and six, they rose to 97%. They examined possible side effects in mothers and infants, but no significant effects were reported. The most common side effect in mothers was local pain. Four infants in the study had symptoms of fever, cough, and congestion but recovered quickly without medical intervention, except for one who was admitted to the hospital due to their age and given antibiotics.

Ramírez et al. [37] showed that participants developed anti-SARS-CoV-2 RBD IgG antibodies in their serum and milk samples. Further, 89% of human milk samples contained IgA, but IgG was reported as the dominant antibody linked to the parenteral route. S1 IgG anti-SARS-CoV-2 was marked as the higher percentage of antibodies in milk, but due to a limitation in the semi-quantitative assessment of S1 IgA levels, they are not comparable. Per the manufacturer, the serum sample levels of IgG anti-SARS-CoV-2 can potentially have neutralizing capabilities. Additionally, a positive correlation between serum and milk antibodies was reported, which leads to the hypothesis that serum antibody levels may be used to predict breast milk antibody levels. The correlation between serum and milk was higher in infants less than a year old, though this observation was based on minimal differences. The study highlighted that a prolonged breastfeeding period might lead to a stronger vaccine-induced antibody concentration in milk, which requires further investigation. In agreement with other studies, no IgM for anti-SARS-CoV-2 was found in 88% of the participants. The change in the composition of breast milk throughout lactation suggests that mechanisms adapt to the infant’s developing immune system.

Rosenberg-Friedman et al. [34] reported a rapid response in antibody production that is highly synchronized between serum and breastmilk, reaching stabilization 14 days after administration of the second dose. The IgG levels peaked at fourteen days after the second dose, and IgA breastmilk levels seem to follow that but are followed by a decrease. All participants exhibited a neutralizing capability via their milk, with IgG being the dominant neutralizing agent, followed by IgA, even though IgA seems to have a higher concentration than IgG.

All serum and milk samples in the study by Schwartz et al. [31] included anti-SARS-CoV-2 IgG, and there was, again, a positive correlation between serum and milk. Their neutralizing function was assessed at 38.3% against the virus. In analyzing the breastfed infants’ oral mucosa, 60% contained IgG anti-SARS-CoV-2. However, no detectable amounts of anti-SARS-CoV-2 antibodies were found in their circulation through dried blood spots analysis.

The absence of IgA in both serum and human milk was reported by Scrimin et al. [36] twenty days after the administration of the second vaccine. This study analyzes the presence of vaccine-generated antibodies from fifteen days and up to four months after the second dose, which could potentially explain the lack of IgA in milk and serum due to its natural kinetics. The study agreed with other studies by Gray et al. [29] and Young et al. [32], which found a dominance of IgG in serum and human milk after vaccination and an absence of IgA after the second dose. The IgA response was the strongest after the first dose and then decreased. As for neutralizing IgG antibodies, their presence was reported in both serum and milk despite the passing of time, which correlates with the results of Estevez Palau et al. [42], where IgG S1 levels were raised after the second dose of the vaccine. Additionally, this study highlighted that 84% of serum samples retain their antibody level rather than milk samples. As for the analysis of adverse effects, no infants were reported with any, which agreed with the current recommendations for the safety of vaccination.

The prospective cohort study by Yeo et al. [38] focused on the neutralizing activity of SARS-CoV-2 antibodies found in breast milk. This study included lactating individuals and their infants and assessed the levels of anti-SARS-CoV-2 isotypes and mRNA found in their breast milk. All lactating participants had neutralizing antibodies in serum before the second dose administration, which increased further after it. Serum antibody levels (IgG, IgA, and IgM) were detected in all individuals and increased even more after the second vaccine. Breast milk samples also contained neutralizing antibodies, specifically after the second dose; all mothers had anti-RBD IgG1 and IgA, along with 88% of those individuals with IgM levels. This study emphasizes the dominance of IgG1 and IgG3 in serum and attributes the low levels of IgM to the preference in class switching to the IgG and IgA isotypes. After administering two vaccine doses, an increase in IgG1 levels in breast milk samples was detected compared to infected individuals, while this study reported IgA dominance. Low numbers of intact mRNA (27%) were detected in serum and (2%) in human milk samples. The serum samples of five infants were analyzed, and none had intact mRNA or neutralizing antibodies.

The study of Young et al. [32] included both vaccinated and infected lactating participants and demonstrated the kinetics and concentrations between the two populations against SARS-CoV-2 antibodies. The vaccinated population exhibited low IgA levels in milk samples in response to vaccination; after the first dose, a rise was reported, but this decreased after the completion of the vaccination. IgG had a more significant and stable response in vaccinated individuals’ milk than in that of the infected group. The breast milk exhibited neutralizing capabilities in both groups. When compared, the infected group is led by an IgA response, and the vaccinated group is dominated by an IgG response, with a high response rate to the first dose which accelerates further after the second. A total of 90 days after the second dose, the IgG levels were higher than they were in the infected group. The IgA response in vaccinated individuals was comparable with that of infected individuals after the first dose, but the response did not last and faltered over time. Both groups were able to generate neutralizing capable antibodies.

Although IgA, specifically secretory IgA (SIgA), is typically the dominant immunoglobulin found in milk and mucosal surfaces, the results of this review indicate that vaccination against COVID-19 in lactating mothers leads to a dominant IgG response. When comparing vaccinated and infected lactating women, we observed a notable difference in the antibody levels between the two groups. Young et al. [32] reported that the infected group was led by IgA and the vaccinated group was led by IgG. The difference between the immune responses was somewhat expected, as the natural infection with the virus elicits a mucosal response and, hence, the generation of IgA, and the vaccine induces a dominant IgG response. The low levels of IgA in the vaccinated group and the preference for the IgG isotype could be due to the intramuscular vaccination route’s exposure to the viral spike protein, favoring antibody class switching to IgG, which becomes the dominant SARS-CoV-2 immunoglobulin in human milk. Low et al. [44] and Juncker et al. [27] report an IgA robust response to the first dose of the vaccine, but the levels begin to decline, and there is no stimulation of a second response from the second dose. Further, in the study of Low et al. [44], the raw data of the spike and RBD IgG and IgA antibodies were converted to picomolar, and a co-dominance between the two antibodies was reported. This highlighted the significance of absolute quantification when comparing IgG and IgA titers. IgA is thought to be important in the initial stages of immune responses, which can explain why IgA levels begin to decline after a certain period.

Furthermore, functional assays were used to assess the effectiveness of the identified antibodies from human milk. Antibodies were tested for the ability to inhibit the virus from entering host cells and infecting them, the ability to bind spike proteins, and the ability to inhibit ACE2 binding. Unfortunately, not all of the selected studies performed neutralization assays, with only eight providing information regarding the functionality of the immunoglobulins. All studies reported the presence of neutralizing antibodies. In Goncalves et al. [20], only one sample exhibited neutralizing capabilities; after purification and concentration, its ability was increased. In the studies performed by Narayanaswamy et al. [26] and Rosenberg-Friedman et al. [34], human milk antibodies were able to neutralize the spike protein and four other variants. Additionally, Perez et al. [33] specified that neutralizing abilities were linked to IgA and IgG, with the latter having the more robust response; neutralizing levels emerged one month post-vaccination and remained above baseline for three months. Neutralizing capabilities are essential in providing answers regarding maternal passive immunity through breastfeeding. Narayanaswamy et al. [26] conducted an analysis of breastfed infant stool samples and found a small number of anti-SARS-CoV-2 antibodies. This reduction can be explained by gut degradation. Schwartz et al. [30] also conducted an analysis of infant oral mucosa and found IgG antibodies against SARS-CoV-2, but none were found in blood samples [31].

### 3.2. Anti-SARS-CoV-2 Antibodies in Infants and Side Effects

The presence of antibodies against SARS-CoV-2 in children/infants was examined in six studies. In detail, a small number of the studies included serum [12,38], stool [26], dried blood spots, and saliva samples [31] taken from the breastfed infants for the detection of anti-SARS-CoV-2-specific antibodies provided from the breast milk of vaccinated mothers. In two studies [29,30], cord blood was collected at delivery.

Yeo et al. [38] reported that none of the five infants included in their study had detectable SARS-CoV-2 anti-spike RBD-specific IgG, IgM, and IgA Abs, as well as vaccine mRNA, in their serum. In their study, Colan and co-workers [12] did not observe severe maternal or infant adverse effects. Interestingly, IgG antibodies against SARS-CoV-2 were not detected in the blood of infants born to postpartum vaccinated mothers, despite high IgG levels being detected in the maternal blood and milk sample. In contrast, infants born to vaccinated mothers who received two vaccine doses during pregnancy had detectable levels of serum anti-SARS-CoV-2 IgG antibodies. Narayanaswamy et al. [26] reported that anti-RBD IgG and anti-RBD IgA were detected in 33% and 30% of infant stool samples, respectively. Interestingly, the levels of anti-RBD antibodies in infant stool correlated with maternal vaccine side effects. Schwartz and co-workers [31] reported the presence of anti-SARS-CoV-2 IgG in the oral mucosa of three out of five breastfed infants; however, IgG antibodies were not detected in their serum samples. Collier et al. [30] evaluated the transplacental transfer of vaccine-elicited and neutralizing antibodies in nine paired maternal and infant cord blood samples. Interestingly, both types of antibodies were detected in infant cord blood samples, suggesting the effective transplacental transfer of antibodies against SARS-CoV-2 from mother to infant. Likewise, Gray et al. [29] observed the presence of both spike- and RBD-specific IgG antibodies in 100% (10 of 10) of umbilical cord blood samples following maternal vaccination. It should be noted that the role of maternal vaccine-elicited IgGs in infants’ immunity remains inconclusive, as these antibodies do not possess a secretory chain and are mainly digested by the infant. The secretory chain is essential in the survival of milk antibodies in the gastric environment and in their proper transfer to the infant (see [9] and the references cited therein).

The side effects of vaccination in infants and/or mothers were assessed in some studies (*n* = 11) by using detailed questionnaires [11,12,26,29,30,35,36,38,40,41,43]. The majority of the studies included in this work reported mild or no side effects in children/infants (Appendix A). However, Perl et al. [11] reported that four infants developed fever after maternal vaccination, while all had symptoms of upper respiratory tract infection (congestion and cough). Overall, local and systemic effects after anti-SARS-CoV-2 vaccination in lactating individuals were documented in 11 studies included in this systematic review, and the analytical results can be seen in Appendix A. The most common side effects included: muscle pain, headaches, fever, chills, and, less commonly, joint pain. The data indicate that lactating women do not suffer from severe vaccine-related reactions. Golan et al. [12] noted that women vaccinated with mRNA-1237 experience more vaccine-induced adverse effects in comparison to BNT-162b2. Narayanaswamy et al.’s [26] interferon-gamma findings can be related to an increase in side effects. As for their breastfed infants, no significant adverse effects were reported. Although in the study of Perl et al. [11], four infants were reported with fever, cough, and congestion, they recovered quickly with no medical intervention. The absence of severe vaccine-related effects in mothers and infants should encourage un-vaccinated mothers who feared the vaccine would have unwanted repercussions for themselves or their children.

The absence of intact mRNA in serum and human milk is essential in highlighting the unstability of vaccine particles; the lack of neutralizing antibodies and mRNA in the serum of infants also proved this. This suggests that there is limited or no exposure or sensitization of the infant by the mRNA in human milk. Low et al. [44] reported minimal mRNA detected in human milk, and these levels can be expected to be digested by the infant’s gut enzymes. Yeo et al. [38] supported the safety of mRNA, where the serum samples of five infants were analyzed, and no mRNA particles or neutralizing antibodies were discovered.

## 4. Conclusions

In conclusion, the evidence from this systematic review indicates that human milk from vaccinated mothers contains anti-SARS-CoV-2 antibodies—specifically, IgG, IgA, and minimal IgM titers. A dominance of IgG is documented in vaccinated mothers, most likely due to the antigenic presentation via intra-muscular injection and the preference for class switching. An increase in IgG levels is typically seen after the second dose, while IgA levels increase after the first dose and then plateau. Antibody neutralization assays showed that these immunoglobulins have the ability to neutralize the spike protein and receptor binding unit of the virus. Additionally, minimal or no amounts of mRNA particles were detected in breast milk. Studies examining the adverse effects on mothers and babies show no significant adverse effects related to COVID-19 vaccination.

Future research should include a larger, more diverse group of participants and incorporate functional assays to examine infant immunological responses. To date, it is unknown whether vaccination provides passive immunity to breastfed infants. The connection between extended breastfeeding and maternal antibody responses should also be explored.

## Figures and Tables

**Figure 1 ijms-24-02957-f001:**
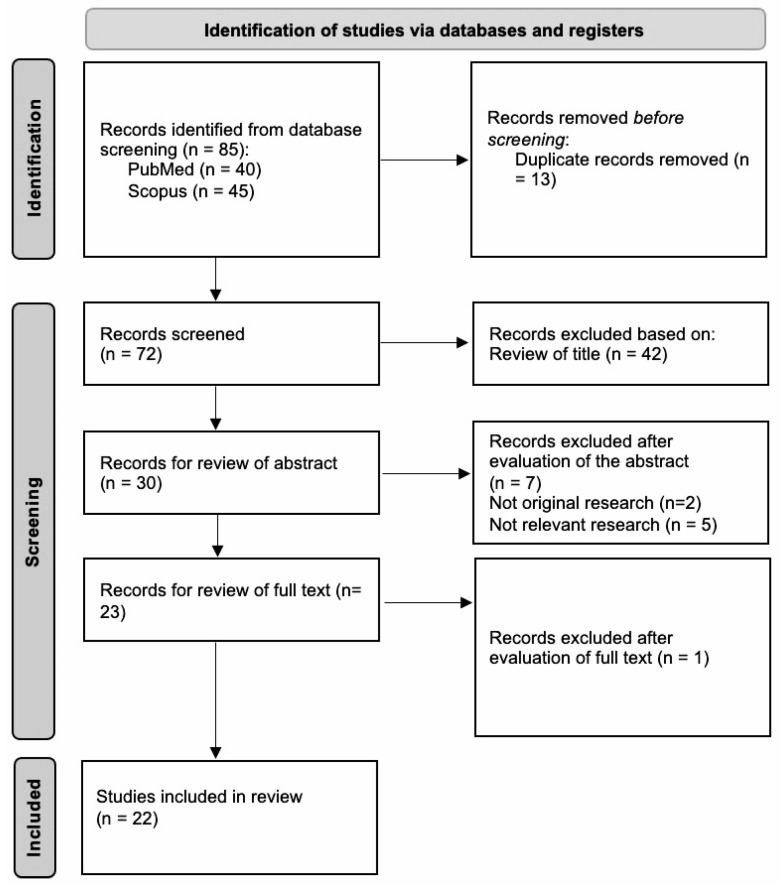
Study selection. Preferred items for Systematic Reviews and Meta-Analyses (PRISMA) flow diagram. Out of 85 identified studies and after the application of the inclusion and exclusion criteria, 22 studies were included in this work.

## Data Availability

Not applicable.

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
