# Peer review of "Detection of SARS-CoV-2–Specific Antibodies in Human Breast Milk and Their Neutralizing Capacity after COVID-19 Vaccination: A Systematic Review"

_ijms, 2023, doi:10.3390/ijms24032957_

Round 1

Reviewer 1 Report

The review is well written and cover most of the aspects regarding the detection of SARS-CoV-2 specific antibodies after covid-19 vaccination. The different scenarios with different type of vaccines are well covered and good space is given to the presence of SARS-CoV-2 specific antibodies in human breast milk. The review summarised the neutralising capacity of SARS-CoV-2 antibodies giving a detailed overview of the state of art of the field. Furthermore, the structure of the review is very well organised.

I strongly recommend the publication of this review in International Journal of Molecular Sciences. 

Author Response

We thank the Reviewer for appreciating our work and recommending the publication of our systematic review.

Reviewer 2 Report

Topic: Detection of SARS-CoV-2–Specific Antibodies in Human Breast Milk and their Neutralizing Capacity After COVID-19 Vaccination: A Systematic Review .

Thank you for asking me to review this paper. The real question I had hoped this review would answer were as follows: Is covid-19 vaccination safe in pregnant women? Is it safe for the unborn child? How much of the antibodies generated by the vaccines is excreted in breastmilk? Do these antibodies retain the ability to neutralize the virus? And finally, how much passive immunity is conferred on the breastfed baby by the vaccinated mother? The paper appears to answer all the above questions with the disappointing exception of the very last one. The paper suggests that covid 19 vaccination is safe for pregnant women, and does not appear to harm the unborn child. It suggests that a considerable quantity of antibodies (a minimal amount of IgA, a considerable amount of IgG, and some IgM) is detectable in the breastmilk of immunized mothers.  It is also suggested that 70% of the IgA are reportedly of the SIgA variety.  It suggests that the secreted antibodies do neutralise the spike protein of the virus. More importantly, the mRNA particles do not appear to cross the blood/milk barrier in worrisome quantities.

It is disappointing that after all we do not know the amount of passive immunity conferred on breastfed babies. This is an area for urgent future research.

Overall, the paper is well written and informative. I recommend acceptance for publication.

Reviewer 3 Report

Thank you for asking me to review this manuscript. It addresses an important question and the introduction clearly lays out the issues. The retrieval of articles is well described and clear for this systematic review. The points made are all valid. 

The main issue is that the discussion seems to comment on each retrieved article in turn and thereby becomes very long. A more concise way would be to structure the discussion in a different way, highlighting the same key findings across all the papers.

Round 2

Reviewer 3 Report

Thank you for your rapid response. I appreciate the comments you have made and that you had already considered presenting the Discussion in a different form. You have made some editorial changes which are acceptable. This remains a relevant manuscript.